# Ecological-Dynamic Approach vs. Traditional Prescriptive Approach in Improving Technical Skills of Young Soccer Players

**DOI:** 10.3390/jfmk9030162

**Published:** 2024-09-14

**Authors:** Giovanni Esposito, Rosario Ceruso, Sara Aliberti, Gaetano Raiola

**Affiliations:** 1Department of Human, Philosophical and Education, University of Salerno, 84084 Fisciano, Italy; saaliberti@unisa.it; 2Research Centre of Physical Education and Exercise, Pegaso University, 80143 Napoli, Italy; rosario.ceruso@univr.it (R.C.); gaetano.raiola@unipegaso.it (G.R.); 3Department of Neuroscience, Biomedicine and Movement, University of Verona, 37129 Verona, Italy

**Keywords:** training, skills, visual occlusion, youth football, evaluation

## Abstract

**Background**: This study contributes to expanding the existing literature on learning technical skills in youth soccer by comparing the effectiveness of different training approaches in the development of passing skills. The ecological-dynamic approach, which emphasizes the continuous and adaptive interaction between the athlete and the environment, is analyzed in comparison to the traditional prescriptive approach, which relies on predefined techniques and exercises. The aim of the study is to determine which of the two approaches is more effective in improving the performance of young soccer players. **Methods**: Thirty players (age 12 ± 1.2 years) were randomly assigned to two groups: the ecological-dynamic group (ECG) and a control group (CON). Both groups underwent an eight-week training program with equal sessions. The ECG group’s training focused on adjusting constraints like the learning environment, game rules, and visual restrictions to boost adaptability and problem-solving skills. The CON group followed a traditional prescriptive approach with specific instructions, goal setting, immediate feedback, and structured exercise progression. Passing abilities were evaluated before and after the program using the Loughborough Soccer Passing Test, with a retention test administered five weeks later. Descriptive statistics, including mean values and percentage improvements, were used. A repeated measures ANOVA compared differences between the groups. **Results**: The analysis revealed a significant Occasion × Group interaction for all performance variables, indicating that the ECG group experienced greater improvements than the CON group. Specifically, the ECG group showed significant reductions in Trial Time (*p* = 0.001, η_p_^2^ = 0.6), Penalty Time (*p* = 0.016, η_p_^2^ = 0.4), and Overall Performance (*p* = 0.011, η_p_^2^ = 0.8) from pre-test to post-test. However, these improvements did not persist into the retention test (*p* = 0.131, *p* = 0.792, and *p* = 0.192, respectively). The CON group also improved significantly in Trial Time (*p* = 0.003), Penalty Time (*p* = 0.002), and Overall Performance (*p* = 0.001) from pre-test to post-test, but with smaller effect sizes and no sustained gains at retention. **Conclusions**: The ecological-dynamic approach (EDG) has proven to be more effective in enhancing passing skills compared to the traditional prescriptive approach (CON). Although both methods led to performance improvements, the EDG group achieved more significant progress.

## 1. Introduction

The acquisition of technical skills in soccer is a complex process that requires a critical evaluation of the training practices adopted [1,2,3]. Research in this area has significant implications for the professional support work of coaches, instructors, and performance analysts, who must constantly define and redefine the criteria for making methodological and didactic choices [4,5]. Consequently, the issue of methodological and didactic choice arises when scientific evidence is conflicting, as in the case of the two main approaches to motor control and learning: the cognitive approach and the ecological-dynamic approach. These approaches explain partially different aspects of motor control and learning, leading to substantially different methodological and didactic implications [6]. The traditional analytical didactic methodology, which has characterized motor and sports training for many years, is primarily based on the cognitive approach. This approach is distinguished by the presentation of sport-specific motor and technical tasks, characterized by a high number of repetitions with executions that are always identical or very similar [7]. Training sessions focus on directive and analytical tasks, aimed at achieving strict adherence to an ideal movement model, through reproduction teaching styles [8]. In this context, the coach plays a central role in the educational process by selecting tasks and defining the methods of execution, quantity, duration, and success criteria [9]. Despite the established practices and theoretical foundations supporting this approach for producing immediate technical results, the reliance on systematic repetition and training standardization tends to overlook the importance of environmental context and situational perception [10,11]. These elements are crucial for developing adaptive technical and decision-making skills [12]. Consequently, this approach promotes only certain learning modes, limiting the discovery of execution variations, problem-solving through practice variability, and adaptation to non-predetermined technical-tactical environmental situations [13]. As a result, the relationship between perception, environment, and problem-solving is constrained, which is vital in real competitive contexts [14].

Conversely, the ecological-dynamic approach offers an alternative perspective on motor skill learning [15]. This approach emphasizes the need for practice contexts that incorporate varying levels of complexity, challenging young practitioners to perform motor patterns in diverse and dynamic settings. Indeed, many of these environmental contexts, which are more meaningful for learning, will never engage the learner in the same way [16]. Learning environments designed according to this methodology utilize affordances to promote desired or developing motor behaviors [17]. The concept of affordances refers to the action opportunities that the environment offers and that athletes perceive in relation to their abilities and the goals they aim to achieve [18]. A soccer player might perceive the opportunity to make a pass or a dribble based on the positions of opponents and teammates. This perception is influenced by their technical skills and the dynamics of the game [19,20]. This direct perception of action opportunities highlights the importance of the interaction between the athlete and the environment, promoting greater adaptability and awareness [21]. In the ecological-dynamic approach, methodological solutions are applied through variability of practice, which is not viewed as a linear obstacle to learning but rather as a cyclical element that justifies and reflects the constraints present [22]. The concept of “constraint” refers to the limitations that influence decision-making and the execution of motor actions [23]. Constraints can be internal, such as the individual’s body dimensions and physical and cognitive abilities, or external, such as the specific demands of the task and the characteristics of the environment [24]. These constraints may also include limitations imposed by the coach and environmental conditions that suggest the best motor solutions. The interaction between these different types of constraints represents an additional procedural element, where the individual seeks to identify optimal motor solutions within the limits imposed by these constraints [25].

Recent studies in the scientific literature have highlighted the distinctive benefits of the ecological-dynamic approach in training young soccer players compared to the traditional approach. Specifically, the ecological-dynamic approach has been associated with improved adaptability and decision-making skills in real game situations, promoting more flexible and situational learning [11,15]. This approach encourages variability in practice and the exploration of different motor solutions, promoting greater environmental awareness and an enhanced ability to respond to dynamic and unpredictable conditions [17]. Moreover, integrating complex and variable practice contexts, typical of the ecological-dynamic approach, has been found to not only improve specific technical skills but also contribute to the development of cognitive skills and tactical creativity in young soccer players [19]. Therefore, the literature supports adopting this approach as an effective strategy for the holistic training of young soccer players. It prepares athletes to face the challenges of modern play with greater adaptability and decisiveness [26,27].

Despite the significant theoretical value of the ecological-dynamic approach, its practical application in youth soccer training remains limited. Many studies have focused on traditional methodologies, which, although effective for the immediate acquisition of technical skills, do not adequately prepare players to handle the variability and complexity of real game situations [28,29,30]. Moreover, existing research has not yet thoroughly explored how the ecological-dynamic approach might influence the learning of technical skills in young soccer players within competitive contexts [31]. This study aims to address this gap by comparing the effectiveness of the ecological-dynamic approach with traditional prescriptive methods in improving passing performance among young soccer players. The hypothesis is that the ecological-dynamic approach will be more effective in enhancing athletes’ technical and decision-making abilities due to its emphasis on variability and situational perception. The anticipated results could provide new insights on how to optimize training practices and better prepare young athletes to face the challenges of real-game scenarios.

## 2. Materials and Methods

### 2.1. Study Participants

Thirty amateur under-13 soccer players from the same youth soccer team in the province of Salerno, competing at the regional level, were assigned to two training groups through simple randomization: one following the ecological-dynamic approach (ECG) and the other following the cognitive approach (CON). Both the ECG (n = 15; mean age 12.6 ± 0.4 years; mean height 170.5 ± 3.0 cm; mean weight 60.6 ± 3.8 kg) and CON (n = 15; mean age 12.7 ± 0.4 years; mean height 170.7 ± 3.1 cm; mean weight 60.4 ± 3.6 kg) groups consisted of an equal number of players in each position (6 defenders, 5 midfielders, and 4 forwards). This sample size was selected due to practical constraints and the availability of players, allowing for adequate analysis within the study’s scope. Participants were recruited from the same team to maintain consistency in training and competition levels. The inclusion criterion was having at least five years of sports experience, while the exclusion criterion included any muscular, tendon, or bone injuries sustained in the previous 12 months. The experimental procedure, risks, and benefits were explained to the parents before participation, and informed consent was obtained from the parents or legal guardians. The study was approved by the PhD Program Board of the University of Salerno (decision no. 16-10-2023), which ensured compliance with ethical standards in the doctoral academic context. Additionally, the study was conducted according to the Declaration of Helsinki, the CONSORT recommendations for non-pharmacological studies [26], and the American Psychological Association’s recommendations for research and publication [32].

### 2.2. Procedure

An experimental field study was conducted in which both groups underwent a differentiated training program lasting eight weeks, consisting of three 90 min training sessions per week, for a total of 24 sessions. Each session consisted of three main phases, each aimed at developing specific technical skills in the players. The sessions were designed to optimize learning and enhance performance through a systematic and integrated approach. The warm-up phase, lasting approximately 15 min, was conducted jointly by both groups and was designed to prepare the young soccer players physically and mentally for training. This initial phase began with mobility and dynamic stretching exercises aimed at improving flexibility and reducing the risk of injury. To ensure an effective warm-up and maintain the players’ focus, practical exercises such as rondo and technical circuits were included. Rondo drills enhance ball control and passing skills in an interactive and stimulating context [33]. Technical circuits, on the other hand, allow for a series of targeted exercises, such as control and dribbling, in a structured and varied manner [34]. The concluding phase of the training, lasting approximately 35 min, involved two simultaneous themed matches of 25 min each, where the experimental group did not compete directly with the control group. This segment provided players with the opportunity to apply the skills they had acquired and refine strategies in a real-game context. During these matches, players engaged in scenarios with specific objectives, such as earning a reward for consecutive passes or for effectively maintaining ball possession. This setup was designed to focus attention on particular aspects of the game and to practice the developed skills, facilitating a practical application of the learned competencies. Each session concluded with 10 min of cool-down exercises, including light jogging, static stretching, and relaxation techniques to aid in recovery and reduce muscle tension. The central phase of the training, lasting approximately 40 min, differed significantly between the two groups.

For the ECG group, this phase focused on situational games designed to enhance participants’ passing skills by simulating realistic game scenarios and refining technical abilities. This phase involved applying specific constraints to emphasize passing, which facilitated the development of these technical skills [35]. Among the primary constraints implemented was limiting the number of touches on the ball to a maximum of three. This rule encouraged players to make quicker and more accurate passes, promoting combined play and reducing reliance on dribbling. Consequently, players were motivated to develop better passing skills and improve their game vision and off-the-ball movement, which are crucial for effective team play. In the context of situational games, one of the main exercises involved using a field with four goals, one of which was covered. This exercise required players to adopt adaptable passing strategies, as they could not use one of the goals and had to focus on the remaining three. This constraint improved the fluidity of combined play and enhanced the players’ ability to create and exploit open spaces for passing. Another exercise involved reducing the size of the field to increase defensive pressure. With tighter spaces, players were compelled to make quick decisions and refine their passing under pressure, simulating game conditions in high-intensity situations. This approach facilitated the development of more precise and timely passing skills, enhancing the players’ ability to manage ball possession in confined areas. Another significant constraint applied during the training phase was visual restriction, designed to improve spatial awareness and movement perception during passing [36]. This constraint was implemented using monocular patches and occlusion goggles. Monocular patches, applied to the dominant eye, forced players to rely on more limited visual information and to use their non-dominant limb more, thereby enhancing their ability to orient themselves and make quick decisions under partial vision conditions [37,38]. Occlusion goggles, which restrict the visual field, helped players focus on essential elements of the game and improve their ability to perceive and react to stimuli [39,40]. These constraints have proven effective in enhancing orientation and rapid decision-making, with a particular focus on passing skills. The implementation of these constraints and techniques aimed to significantly improve players’ passing abilities, ensuring greater accuracy and fluidity in their on-field performance. The coach adopted a less direct role, providing only non-verbal feedback such as visual cues and general guidance. Athletes had the opportunity to observe performance models through video tutorials, which facilitated their reflection and self-correction. These non-prescriptive videos allowed athletes to review their performance, identify areas for improvement, and make adjustments independently [41]. This approach encouraged self-processing and self-regulation, supporting skill development without the need for direct intervention from the coach.

The CON group followed the cognitive-based training program during the central phase of the session, based on the guidelines of the Territorial Development Program coordinated by the Youth and Scholastic Sector of the Italian Football Federation [42]. In this phase, the training focused on analytical drills, progressing from simplified exercises to more complex exercises. The initial drills were designed to be less complex, facilitating the acquisition of basic technical skills. Players were given clear instructions on how to perform the passes, and each drill had specific performance goals. Immediate formative feedback was provided after each drill. Coaches intervened promptly when players failed to complete the task, allowing them to repeat the same or similar gesture. Players were motivated and encouraged through verbal support. Although the drills had limited representation of real-game situations, they were useful for developing the players’ technical skills. The graphic representation of the training protocol is shown in Table 1.

### 2.3. Data Collection and Measurement

In this study, a standardized and validated assessment tool, the Loughborough Soccer Passing Test [43], was employed to evaluate the effectiveness of the ECG and CON training protocols on technical passing skills. The test was administered at three separate times: before the start of the training program (pre-test), immediately after the completion of the training program (post-test), and five weeks later (retention test). This assessment aimed to measure the accuracy of players in executing repeated passes under time constraints. Participants who were already familiar with the test focused on performing precise passes within a designated area, with particular emphasis placed on spatial and temporal skills. The test was conducted within a rectangular area measuring 12 m by 9.5 m, which contained two smaller rectangles: one measuring 4 m by 2.5 m and the other 2.5 m by 1 m. Additionally, there was a 0.75 m-wide corridor surrounding the smaller rectangle. Cones marked the corners of the central rectangles and the center of the smaller rectangle. Rectangles measuring 2.5 m by 30 cm were placed along the outer perimeter, with a 1 m-wide colored section in the center. The target was a metal plate 30 cm wide, positioned within a target area measuring 60 cm by 30 cm.

Players were required to execute 16 passes: 8 towards the long sides of the outer rectangle (red and white) and 8 towards the short sides (blue and green), aiming to hit the metal plate. The test involved two operators: one to manage the timing and the other to call out the colors of the targets in a random sequence. The ball had to pass through the corridor between the inner rectangles, bounce off the colored rectangles, and return to the central cone before being kicked towards the next target. The test duration was 43 s, starting when the player entered the corridor. Performance was measured by an experienced observer using a stopwatch. The penalties and bonuses applied were as follows: 5 s added for a missed shot or a shot on the wrong rectangle; 3 s for hand contact; 3 s for missing the target area; 2 s for failing to kick within the corridor 2 s for hitting a cone; and 1 s for each second exceeding 43 s. Additionally, players received a 1 s bonus for each successful hit on the metal plate.

Players’ performances were evaluated based on three variables: the total time to complete the 16 passes (Trial Time), the time adjusted for penalties and bonuses (Penalty Time), and the total time after applying penalties and bonuses (Overall Performance). Each participant was allowed two attempts per test, with only their best result being considered. Testing conditions were standardized across all evaluations to ensure consistency and minimize potential influencing variables. During the intervention period, all testing sessions were recorded using a GoPro Hero8 Black^®^ (GoPro, San Mateo, CA, USA).

The technical passing test provides a direct indication of how different training methods impact players’ specific skills. While it does not fully replicate real game conditions, it offers valuable data on the acquisition and refinement of technical skills. This approach allows for the assessment of whether training based on the ecological-dynamic approach (ECG), which emphasizes variability and representativeness, results in tangible improvements in passing skills beyond the complex dynamics of real gameplay. Furthermore, the standardized passing test clarifies how and to what extent ECG training can affect isolated technical performance, a crucial aspect for understanding whether more representative and variable training methodologies can also be effective outside of real game contexts. By providing new insights into the effectiveness of different training approaches, the study contributes to the academic debate and enhances the understanding of learning dynamics in soccer [44,45].

### 2.4. Statistical Analysis

The data reported in descriptive form are expressed with mean (M) and standard deviation values (SD). The normality of the data distribution was verified using the Shapiro–Wilk test, while the homogeneity of the variances was verified with the Levene test. An independent-sample *t*-test was performed to assume non-significant differences between values before the training program. Separate 2 (Group) × 3 (test Occasions) repeated measure ANOVA were conducted to examine the interventions interactions and differences within and between the groups. In the case of statistically significant interactions or main effects, Bonferroni post hoc tests were conducted to compare the pre-test, post-test, and retention conditions. The level of statistical significance was defined at *p* ≤ 0.05, with a confidence interval for differences set at 95%. For the evaluation of the effect sizes, the benchmarks were set as follows (d = 0.2, small; d = 0.5, medium; d = 0.8, large) [46]. All statistical tests were processed using IBM SPSS (version 22; IBM, Armonk, NY, USA).

## 3. Results

Initial statistical analysis of the training groups, using an independent-sample *t*-test on pre-test data, revealed no statistically significant differences between the groups in Trial (*p* = 0.208), Penalty Time (*p* = 0.264), and Overall Performance (*p* = 0.546). Table 2 shows the average (±SD) time (s) needed to complete the Loughborough Soccer Passing Test for the two treatment groups.

Table 3 presents the results of the two-way ANOVA, examining the effects of Occasion, Group, and their interaction on the measured variables. Table 4 provides the results of post hoc comparisons for these performance variables.

For the “Trial” variable, the main effect of Occasion was significant (*p* = 0.001, η_p_^2^ = 0.9, 95% CI [0.8; 0.9]), indicating that performance varied significantly across different test Occasions. The main effect of Group was also significant (*p* = 0.001, η_p_^2^ = 0.9, 95% CI [0.8; 0.9]), suggesting that there were differences between ECG and CON groups. The analysis further revealed a significant interaction between Group (ECG and CON) and Occasion (pre-test, post-test, retention), with a *p*-value of 0.001 and a medium effect size (η_p_^2^ = 0.6, 95% CI [0.3; 0.7]). Post hoc comparisons between the pre-test and post-test showed a significant improvement in performance (*p* = 0.001, η_p_^2^ = 0.8, 95% CI [0.7; 0.9]), indicating a high effect size. However, comparisons between the post-test and retention did not reveal significant differences (*p* = 0.131, η_p_^2^ = 0.07, 95% CI [0.1; 0.2]), suggesting that the improvements did not persist over time. For the variable “Penalty Time (PT)”, the main effect of Occasion was significant (*p* = 0.001, η_p_^2^ = 0.9, 95% CI [0.8; 0.9]), as was the main effect of Group (*p* = 0.001, η_p_^2^ = 0.9, 95% CI [0.8; 0.9]). Additionally, the analysis revealed a significant interaction between Group and Occasion (*p* = 0.016, η_p_^2^ = 0.4, 95% CI [0.1; 0.5]). Post hoc comparisons between the pre-test and post-test showed a significant reduction in Penalty Time (*p* = 0.001, η_p_^2^ = 0.4, 95% CI [0.1; 0.6]), reflecting a moderate effect size. However, similarly to the “Trial” variable, the comparison between the post-test and retention was not significant (*p* = 0.792, η_p_^2^ = 0.002, 95% CI [0.0; 0.1]). Finally, for the variable “Overall Performance (OP)”, the main effect of Occasion was significant (*p* = 0.001, ηp^2^ = 0.9, 95% CI [0.9; 0.9]), as well as the main effect of Group (*p* = 0.001, ηp^2^ = 0.9, 95% CI [0.9; 0.9]). The interaction between Group and Occasions was also significant (*p* = 0.011, η_p_^2^ = 0.8, 95% CI [0.6; 0.9]). Post hoc comparisons between the pre-test and post-test showed significant improvements (*p* = 0.001, η_p_^2^ = 0.8, 95% CI [0.7; 0.9]), indicating a considerable effect. However, the comparison between post-test and retention was not significant (*p* = 0.192, η_p_^2^ = 0.06, 95% CI [0.1; 0.2]), suggesting that the improvements were not sustained over the long term. These results indicate that, although there were significant improvements between the pre-test and post-test for all performance variables, the effects of these improvements tend to diminish over time, as evidenced by the non-significant comparisons between post-test and retention.

To assess changes in performance over time, intra-group comparisons are summarized in Table 5.

In the ECG group, significant improvements were observed in all three performance variables from the pre-test to the post-test. For the average time to complete the 16 passes (Trial), there was a notable improvement with a *p*-value of 0.001, a medium effect size (d = 0.6), and a confidence interval of [7.3; 15.8]. However, the change from post-test to retention was not significant (*p* = 0.521), with a small effect size (d = −1.5) and a confidence interval of [−2.2; −0.7]. Regarding penalty time (PT), there was a significant reduction from the pre-test to the post-test (*p* = 0.001, d = 0.5), with a confidence interval of [2.9; 6.6]. The change from post-test to retention was not significant (*p* = 0.214), with a large effect size (d = −3.2) and a confidence interval of [−4.4; −1.9]. For overall performance (OP), a significant improvement was seen from the pre-test to the post-test (*p* = 0.001, d = 0.8), with a confidence interval of [7.4; 16.1]. The change from post-test to retention was not significant (*p* = 0.146), with a large effect size (d = −2.2) and a confidence interval of [−3.2; −1.2].

In the CON group, improvements were also significant across all variables from the pre-test to the post-test. For the average time to complete the 16 passes (Trial), the significant change (*p* = 0.003) had a small effect size (d = 0.3) and a confidence interval of [10.2; 22.0]. The change from post-test to retention was not significant (*p* = 0.106), with a small effect size (d = −0.5) and a confidence interval of [−1.1; −0.2]. For penalty time (PT), there was a significant reduction from the pre-test to the post-test (p = 0.002, d = 0.4), with a confidence interval of [2.5; 5.8]. The change from post-test to retention was not significant (*p* = 0.139), with a large effect size (d = −2.1) and a confidence interval of [−3.1; −1.2]. For Overall Performance (OP), the significant improvement from the pre-test to the post-test (*p* = 0.001, d = 0.4) had a confidence interval of [9.6; 20.9]. The change from post-test to retention was not significant (*p* = 0.126), with a medium effect size (d = −1.1) and a confidence interval of [−4.0; −0.1].

## 4. Discussion

This study aimed to identify the most effective learning approach for developing the technical skills of young soccer players by comparing the ecological-dynamic approach with the cognitive-prescriptive approach in passing performance. The ECG group followed a protocol based on the principles of the ecological-dynamic approach, which altered the game constraints to promote the development of passing skills. The training sessions included specific constraints, such as limiting the number of touches with the ball and using occlusion goggles, designed to stimulate the players’ adaptation and decision-making abilities in variable game situations. This approach promoted greater exploration of game strategies and improved skill transferability [47,48]. In contrast, the CON group followed a traditional approach based on the cognitive approach with targeted and repeated exercises [13], designed to improve passing skills through a gradual progression of difficulty in a controlled environment.

The innovation of this study lies in the use of a field-based technical test, which could have favored the CON group given their greater affinity with the repetitive exercises of that approach. However, the results challenge the conventional idea that a high number of decontextualized repetitions is necessary to develop technical skills [49]. Although real performance in a match can provide valuable insights, it is influenced by numerous external variables, such as opponent tactics, environmental conditions, and the player’s psychophysical state [50]. These factors make it very difficult to isolate variables for measurement. Consequently, advancing knowledge on the specific impact of one training approach compared to another is challenging, as on-field performance depends on many uncontrollable factors [51]. Therefore, the standardized test used in our study allows for a more controlled and precise evaluation of the effects of training approaches, reducing the influence of external variables.

The research suggests that the ecological-dynamic approach, with its emphasis on variable and representative contexts, may offer a more effective method for improving not only specific performance but also skill transferability. Although no significant differences emerged between the post-test and the retention test for both groups, the ECG group achieved more marked improvements compared to the CON group in all measured variables (Trial, PT, and GP) from the pre-test to the post-test. This suggests a greater effectiveness of the ecological-dynamic approach compared to the cognitive-prescriptive approach, contradicting expectations based on the principle of practice specificity, which argues that highly specific training should lead to optimal improvements in corresponding test conditions [52,53]. The ecological-dynamic approach, characterized by greater variability and dynamism, seems to facilitate the development of more adaptable and transferable skills. This could explain why the ECG group, despite training that was less focused on exactly replicating the final test conditions compared to the CON group, achieved superior results. The variability introduced during training may have stimulated a more flexible type of learning, enhancing the participants’ ability to adapt to situations different from those specifically trained for. The short-term improvement observed in the CON group might result from highly specific and targeted learning, which may not adequately prepare participants to handle variable and less predictable situations. The findings indicate that the design of training programs should incorporate elements of variability and dynamism to optimize not only immediate improvements but also their persistence and applicability in different contexts.

These results align with the findings of Sannicandro et al. [28], which showed that a protocol based on the ecological-dynamic approach can enhance the physical and technical performance of young soccer players more effectively than traditional prescriptive methods. The ecological-dynamic approach, characterized by the manipulation of game constraints and the variability of stimuli, appears to promote the development of motor skills that can be easily adapted and transferred to different game situations [17]. This supports the idea that exposure to variable contexts during training can foster more flexible learning, enhancing the ability to apply acquired skills across a variety of different scenarios [54].

Similarly, Buszard et al. [55] highlighted that the ecological-dynamic approach is more effective in facilitating motor skill learning compared to traditional methods, which rely on general and global instructions provided by instructors. The modifications to task constraints, environment, and game rules allow children to explore and adapt their actions in variable and more representative contexts. In contrast, the cognitive-prescriptive approach, while effective in the short-term due to its gradual progression, may not adequately prepare players to handle variable and unpredictable game situations [12,56]. The results obtained with the cognitive-prescriptive approach show rapid improvements, but these gains are often limited and do not necessarily translate into greater adaptability to complex and dynamic game contexts. Training program design should therefore incorporate elements of variability and dynamism to maximize not only immediate improvements but also their relevance and sustainability in the long term.

This study has several limitations that should be carefully considered. Although the use of a technical test was innovative, it does not fully align with the principles of the ecological-dynamic approach. Specifically, the test created a static and simplified environment that does not accurately reflect the complexity and variability of real game situations. The ecological-dynamic approach stresses that skill assessments should be conducted in contexts that closely mimic actual game conditions, where technical and tactical components are integrated. Additionally, the sample examined, which consists of under-13 players from a single soccer academy, is limited in both size and diversity. This limitation may impact the generalizability of the results to broader contexts or different groups of athletes. For future research, it would be advantageous to include additional measures that offer a more comprehensive evaluation of performance. Such measures could include qualitative analyses of tactical decisions and strategies, as well as assessments of athlete perceptions [57]. From a practical perspective, the study’s findings suggest that training programs should integrate elements of variability and dynamic contexts, even in technical assessments. Incorporating more representative and adaptable training scenarios could enhance the transferability and adaptability of players’ skills, thereby better preparing them for the complexities encountered in competitive play. This approach may lead to more effective skill development and improved performance in real-game situations.

## 5. Conclusions

The results of this study indicate that the ecological-dynamic approach is more effective than the cognitive-prescriptive approach for enhancing both specific performance and skill transferability in soccer training. The ecological-dynamic approach, with its focus on variable and representative contexts, led to greater improvements in performance measures (Trial, PT, and GP) from the pre-test to the post-test. This approach’s emphasis on variability and dynamism appears to foster more adaptable and transferable skills, allowing players to better handle diverse and complex game situations. These findings suggest that incorporating elements of variability and dynamism into training programs can optimize not only immediate performance gains but also their relevance and persistence across different contexts. However, the study has some limitations, including the small sample size, the limited duration of the intervention, and the specificity of the test used. These factors highlight the need for further research to confirm and extend these conclusions in broader and more diverse sports contexts.

## Figures and Tables

**Table 1 jfmk-09-00162-t001:** Training protocol performed by the two groups.

ECG Group	CON Group
Frequency: Three weekly sessions for eight weeks	
Session duration: Approximately 90 min	
Dynamic warm-up: Start with a 15 min dynamic warm-up, including 5 min of dynamic stretching, 5 min of rondo exercises (2 × 2 min), and 5 min of technical station circuits (5 stations, 1 min each).	
Central phase: (40 min). Focuses on situational games with specific constraints to enhance passing skills. Exercises include: Limiting ball touches to a maximum of three (5 min) Using a field with four goals (one covered) Reducing field size for increased defensive pressure (10 min) Applying visual restrictions with monocular patches and occlusion goggles (15 min) Non-verbal feedback including visual cues and general guidance, with performance models provided through video tutorials for self-reflection and self-correction (10 min)	Central phase: (40 min). Focuses on analytical drills progressing from simplified to complex exercises. Includes: Clear instructions for pass execution Specific performance goals for each drill Immediate formative feedback Verbal support and encouragement
Concluding Phase: Themed matches of 25 min each; allows players to apply and refine acquired skills in a real-game context with specific objectives. Cool-down: 10 min. Includes light jogging, static stretching, and relaxation techniques to aid in recovery and reduce muscle tension.	

**Table 2 jfmk-09-00162-t002:** Average (±SD) time (s) needed to complete the Loughborough Soccer Passing Test for the two treatment groups.

	ECG (n = 15)			CON (n = 15)		
Variable	Pre-Test	Post-Test	Retention	Pre-Test	Post-Test	Retention
Trial (s)	49.9 ± 0.5	43.0 ± 0.6	43.7 ± 0.5	50.3 ± 0.6	46.9 ± 0.6	48.9 ± 0.5
PT (s)	10.5 ± 0.5	7.5 ± 0.3	8.1 ± 0.3	11.2 ± 0.2	9.5 ± 0.2	10.2 ± 0.2
OP (s)	59.9 ± 0.5	50.3 ± 0.6	50.8 ± 0.6	60.2 ± 0.6	56.3 ± 0.6	59.1 ± 0.6

Note: PT: Penalty time; OP: Overall Performance; ECG: Ecological-dynamic group; CON: Control group.

**Table 3 jfmk-09-00162-t003:** Results of two-way ANOVA for the effects of Occasion, Group, and their interaction on the performance variables.

Variable	Occasion		Group		Occasion × Group Interaction	
	*p*	η_p_^2^ (95% CIs)	*p*	η_p_^2^ (95% CIs)	*p*	η_p_^2^ (95% CIs)
Trial (s)	0.001 *	0.9 (0.8; 0.9)	0.001 *	0.9 (0.8; 0.9)	0.001 *	0.6 (0.3; 0.7)
PT (s)	0.001 *	0.9 (0.8; 0.9)	0.001 *	0.9 (0.8; 0.9)	0.016 *	0.4 (0.1; 0.5)
OP (s)	0.001 *	0.9 (0.9; 0.9)	0.001 *	0.9 (0.9;0.9)	0.011 *	0.8 (0.6; 0.9)

* Result significantly different from pre-test result (*p* < 0.05).

**Table 4 jfmk-09-00162-t004:** Post hoc comparisons of performance variables.

Variable	Pre-Test vs. Post-Test		Post-Test vs. Retention	
	*p*	η_p_^2^ (95% CIs)	*p*	η_p_^2^ (95% CIs)
Trial (s)	0.001 *	0.8 (0.7; 0.9)	0.131	0.07 (0.1; 0.2)
PT (s)	0.001 *	0.4 (0.1; 0.6)	0.792	0.002 (0.0; 0.1)
OP (s)	0.001 *	0.8 (0.7; 0.9)	0.192	0.06 (0.1; 0.2)

* Result significantly different from pre-test result (*p* < 0.05).

**Table 5 jfmk-09-00162-t005:** Intra-group comparisons (ECG vs. CON) of performance variables.

ECG (n = 15)					CON (n = 15)			
Variable	Pre-Test vs. Post-Test		Post-Test vs. Retention		Pre-Test vs. Post-Test		Post-Test vs. Retention	
	*p*	d (95% CIs)	*p*	d (95% CIs)	*p*	d (95% CIs)	*p*	d (95% CIs)
Trial (s)	0.001 *	0.6 (7.3; 15.8)	0.521	−1.5 (−2.2; −0.7)	0.003 *	0.3 (10.2; 22.0)	0.106	−0.5 (−1.1; 0.2)
PT (s)	0.001 *	0.5 (2.9; 6.6)	0.214	−3.2 (−4.4; −1.9)	0.002 *	0.4 (2.5; 5.8)	0.139	−2.1 (−3.1; −1.2)
OP (s)	0.001 *	0.8 (7.4; 16.1)	0.146	−2.2 (−3.2; −1.2)	0.001 *	0.4 (9.6; 20.9)	0.126	−1.1 (−4.0; −0.1)

* Result significantly different from pre-test result (*p* < 0.05).

## Data Availability

All data generated or analysed during this study have been included within the manuscript.

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
