# Peer review of "Ecological-Dynamic Approach vs. Traditional Prescriptive Approach in Improving Technical Skills of Young Soccer Players"

_jfmk, 2024, doi:10.3390/jfmk9030162_

Round 1

Reviewer 1 Report

Comments and Suggestions for Authors

Initially, he congratulated the authors for their valuable research work. I am also grateful for having been able to read the document. The following are a series of comments that seek to improve the authors' manuscript. All comments can also be found in the attached pdf document.

Title

It is suggested to review the title

Ecological-Dynamic Approach vs. Traditional Prescriptive Aproach in Improving Technical Skills of Young Soccer Players.

Abstract

It would be important to add an initial paragraph in the introduction that evidences the contribution of this study.

The contribution of this conclusion is not clear. Revise and restructure to respond to the objective of the study when comparing two types of approaches.

Introduction

The introduction does not consider an explanation of what the scientific literature says when using this type of approach (Ecological-dynamic approach versus traditional prescriptive approach) in the training of young soccer players. It is suggested to revise and add a paragraph explaining the effects that this type of approach produces in these population groups.

It is not clear whether this last section of the introduction is part of the procedure or of the methodology. In any case, in addition to the theoretical foundation, other references and studies are needed to highlight the importance of this study. It is suggested to review and restructure.

 Methods

They report that they were U-14 athletes, although the average age of both groups does not exceed 12.6 years.

It would be necessary and ideal to have the approval of the ethics committee for this type of studies. It would also be necessary to review the regulations that explain that all the procedures were established following the guidelines of the Declaration of Helsinki or any other guidelines on the evaluation in human beings.

He considered that the procedure is very well written. However, it is important that they can add a figure that schematically represents the methodological process they carried out.

It is advisable to specify the times for each of the exercises performed. This would make it possible to understand the changes established between the pre-test and the post-test.

Results

He considered that the results could be improved in their presentation. There, it is important that they can add figures that help to schematically understand the changes developed between the pre-test and post-test.

It is suggested that a title be added to the description of the table.

Another important contribution is that the significance values found different from p≤0.05 should be appended within the same table. Also add in the table the effect size values.

This can help to avoid having to explain whether each variable that was significant had an effect size (small, medium, large). In that sense, the values that define what each effect size value means are already explained.

Discussion

One of the major difficulties I find in the paper has to do with the discussion. There is no development where other findings that have used the dynamic-ecological approach and the traditional prescriptive approach are considered. It is suggested that the discussion be reviewed in detail and reconstructed in order to give greater scientific solidity to the conceptual foundation.

Conclusions

It is suggested that the scope of the conclusions of the study be revised. Within what is written there is no clarity on what resulted from the research process. There is no discussion of the characteristics of the approaches or what type of variability produces significant improvements.

On the other hand, it is suggested to include a section on the main limitations of the study, such as the sport level, etc.

Another suggested section has to do with the practical applications of the findings and the future perspectives of the study. The research approach should allow the findings to have a practical contribution. Therefore, it is important for the research group to present these contributions.

References

Review the journal regulations to adjust all the standards.

Reviewer 2 Report

Comments and Suggestions for Authors

This study compares the effects of two training approaches—the ecological-dynamic approach and the prescriptive approach on improving soccer passing performance in teenagers. The results indicate that the ecological-dynamic approach is more effective, even in standard passing tasks.

There are two major concerns with this manuscript:

  1. Statistical Analysis: The study used a 2-way repeated measures ANOVA to compare the groups. However, the results presented appear to focus on a two-groups comparison, reporting the main effects or interactions is missing. 

  2. Ethics Approval: The study lacks ethics/IRB approval. According to the journal's guidelines, IRB approval is required for studies involving human . While the authors mention that IRB approval can be waived in their country for this type of research, it appears that the regulations they cite pertain primarily to data protection rather than research as a whole. I recommend that the authors provide the relevant sections of the guidelines for further review.

Other Comments:

  • Abstract: Please include a brief description of the ecological-dynamic approach. Additionally, ensure that subheadings are included as per the submission guidelines.

  • Methods: Please justify the sample size and clarify how participants were recruited.

  • Discussion: It would be helpful if the authors could explain why real performance in a match is not a suitable or feasible measure for assessing the effectiveness of the new approach.

Round 2

Reviewer 1 Report

Comments and Suggestions for Authors

I thank the authors for the research exercise.

They have managed to resolve all the suggestions made and their publication is considered in the current state.

Author Response

Thank you for your positive feedback on our research. Your valuable suggestions greatly contributed to the improvement of our work.

Best regards

Reviewer 2 Report

Comments and Suggestions for Authors

Regarding the ethics approval, I accept the explanation and relevant policy provided by the authors.

For the statistical analysis, the authors now present the results of a 2-way ANOVA. The reporting remains incomplete. The main effects of occasion and group effects are missing. Additionally, the order of presentation should be revised as the ANOVA results (Table 4) should logically be presented before the intra-group comparisons (Table 3).

Furthermore, has there been any adjustment of p-values for multiple comparisons?

For the newly revised conclusion, the statement in line 437, 'In contrast, while the cognitive-prescriptive approach achieved short-term improvements, it did not prepare participants as effectively for variable and unpredictable scenarios,' cannot be supported by the findings as the study did not perform any assessment of unpredictable scenarios/real performance.

Lastly, the title of Table 4 should be revised to "Occasion x Group" instead of "Moment × Group."

Round 3

Reviewer 2 Report

Comments and Suggestions for Authors

The authors have addressed all my comments.